



# An automated system for trace gas flux measurements from plant foliage and other plant compartments

Lukas Kohl[1,2,⋆], Markku Koskinen[1,2,⋆], Tatu Polvinen[1,2], Salla Tenhovirta[1,2], Kaisa Rissanen[3],
Marjo Patama[1,2], Alessandro Zanetti[4,2], and Mari Pihlatie[1,2,5]

[1]University of Helsinki, Department of Agriculture, Environmental Soil Science Unit, Viikinkaari 9, 00790 Helsinki, Finland
[2]University of Helsinki, Institute for Atmosphere and Earth System Research / Forest Research, Viikinkaari 9, 00790 Helsinki, Finland
[3]Département des sciences biologiques, Université du Québec à Montréal, 141 Avenue du Président-Kennedy, Montreal, QC H2X 1Y4, Canada
[4]University of Helsinki, Department of Forestry, Ladokartanonkaari 7, 00790 Helsinki, Finland
[5]University of Helsinki Viikki Plant Science Center (VIPS), Viikinkaari 9, 00790 Helsinki, Finland
⋆These authors contributed equally to this work.

**Correspondence:** Lukas Kohl (lukas.kohl@helsinki.fi); Markku Koskinen (markku.koskinen@helsinki.fi)

**Abstract.** Plant shoots can act as sources or sinks of trace gases including methane and nitrous oxide. Accurate measurements of these trace gas fluxes require enclosing of shoots in closed non-steady state chambers. Due to plant physiological activity, this type of enclosures, however, lead to $CO_2$ depletion in the enclosed air volume, condensation of transpired water, and warming of the enclosures exposed to sunlight, all of which may bias the flux measurements. Here, we present *PlasTraGAS*, a

5 novel measurement system designed for continuous and automated measurements of trace gas and volatile organic compound (VOC) fluxes from plant shoots. The system uses transparent shoot enclosures equipped with Peltier cooling elements and automatically replaces fixated $CO_2$, and removes transpired water from the enclosure. The system is designed for measuring trace gas fluxes over extended periods, capturing diurnal and seasonal variations and linking trace gas exchange to plant physiological functioning and environmental drivers. Initial measurements show daytime $CH_4$ emissions two pine shoots of

10 0.056 and 0.089 $\mathrm{nmol\,g^{-1}}$ foliage d.w. $\mathrm{h^{-1}}$ or 7.80 and 13.1 $\mathrm{nmol\,m^{-2}\,h^{-1}}$. Simultaneously measured $CO_2$ uptake rates were 9.2 and 7.6 $\mathrm{mmol\,m^{-2}\,h^{-1}}$ and transpiration rates of 1.24 and 0.90 $\mathrm{mol\,m^{-2}\,h^{-1}}$. Concurrent measurement of VOC emissions demonstrated that potential effects of spectral interferences on $CH_4$ flux measurements were at least ten-fold smaller than the measured $CH_4$ fluxes. Overall, this new system solves multiple technical problems that so far prevented automated plant shoot trace gas flux measurements, and holds the potential for providing important new insights into the role of plant foliage in the

15 global $CH_4$ and $N_2O$ cycles.



## 1 Introduction

Plants were recently recognized as potential sources and sinks of atmospheric trace gases including the greenhouse gases methane ($CH_4$) and nitrous oxide ($N_2O$) (e.g. Keppler et al., 2006; Pangala et al., 2015; Machacova et al., 2016; Carmichael et al., 2014; Machacova et al., 2019). Measurements of the $CH_4$ and $N_2O$ exchange between plants and the atmosphere, however, so far remain mostly limited to stem surface fluxes (**?**Covey and Megonigal, 2019), where recent advances in measurement techniques enabled continuous measurements of trace gas fluxes by automated chamber systems (Barba et al., 2019). While the $CO_2$ exchange from plant shoots has been measured for more than a century, few direct measurements of the $CH_4$ and $N_2O$ exchange of plant shoots and/or foliage have been reported thus far (Machacova et al., 2016; Sundqvist et al., 2012; Takahashi et al., 2012). In particular, no continuous measurements of tree shoot $CH_4$ or $N_2O$ exchange have yet been conducted. This lack of available shoot flux data stands in contrast to reports of $CH_4$ and $N_2O$ emissions from plant foliage under laboratory conditions and widespread speculation about their role in the global $CH_4$ and $N_2O$ cycles (e.g. Keppler et al., 2006; Lenhart et al., 2018).

This data gap likely results from the high degree of technical difficulty associated with leaf-level trace gas flux measurements. Due to the small $CH_4$ and $N_2O$ exchange rates at leaf surfaces relative to their atmospheric background mixing ratio, fluxes of these gases can only be measured by static (i.e., non-flow-through, non-steady state) chamber techniques. In such measurements, a plant shoot is enclosed and the change in the trace gas mixing ratio over time is monitored in the enclosed air (e.g. Pihlatie et al., 2005, 2013). Such non-steady-state measurements, however, are impeded by other changes to the chemical and physical properties of the enclosure air volume. Plant shoots transpire water ($H_2O$), fixate carbon dioxide ($CO_2$), and emit volatile organic compounds (VOCs) at rates much higher than trace gases fluxes (e.g. Seco et al., 2007). This leads to the rapidly accumulation of $H_2O$ and VOCs and the depletion of $CO_2$ in the enclosed air volume. In addition, solar irradiation heats the enclosed space and temperatures 10 degrees above ambient conditions have been reported even in large soil surface enclosures (Koskinen et al., 2014).

When measured concurrently with trace gas, fluxes of $CO_2$, $H_2O$, and VOCs can provide additional information on the mechanisms that control plant trace gas emissions. Water and $CO_2$ fluxes allow to quantify the gas conductivity of the leaf surface (i.e., stomatal conductance), and leaf metabolic activities (photosynthesis and respiration rates), respectively. Simultaneous measurements of VOC fluxes allow to assess the potential links between $CH_4$ and co-produced reactive compounds in plant foliage, and thus help identify the source process of $CH_4$ emissions (Benzing et al., 2017). In addition, VOC emissions may cause spectral interferences on trace gas analyser (Kohl et al., 2019b). Monitoring VOC fluxes concurrently with trace gas fluxes can therefore help ensure the validity of trace gas flux measurements.

Continuous, automated, and frequent measurements of plant shoot trace gas exchange will lead to important insights into the the basic mechanisms of plant-atmosphere interactions and the role of vegetation in the global cycles of $CH_4$, $N_2O$, and other trace gases. Realizing this potential, however, requires a solution to the above-mentioned technical challenges. Here, we present *PlaSTraGAS* (PLAnt Shoot TRAce Gas exchange Analyser System), a measurement system capable of measuring trace gas exchange at plant shoots while regulating temperature, humidity, and $CO_2$ mixing ratios in the shoot enclosure.





We designed the system as a modular and adaptable setup to different measurement projects, and so far have constructed two implementations optimized for distinct measurement needs (Fig. 1). The first implementation, designated *PlaSTraGAS-clim2*, is connected to two chambers placed inside a climate controlled plant growth cabinet and is currently used to measure shoot $CH_4$ emissions and root-to-shoot $CH_4$ transport under controlled environmental conditions (Fig 1a). The second implementation, named *PlasTraGAS-gh7*, fits up to seven shoot chambers and is currently used for treatment-control experiments with tree seedlings in a greenhouse compartment (Fig. 1b).

Both systems are capable of (1) temperature control (cooling) of each shoot chamber; (2) automated static chamber (i.e., closed-loop) trace gas exchange measurements of inert gases (e.g. $CH_4$, $N_2O$) with autonomous $CO_2$ addition and removal of excess humidity; (3) dynamic chamber (i.e., flow-through, steady-state) measurements of $CO_2$, $H_2O$, and VOC fluxes; and (4) flushing of shoot chamber with ambient air between the measurements, and 5) recording of temperature and photosynthetically active radiation (PAR) from the chambers.

In this publication, we describe the setup of the two systems and provide results from initial tests and a validation experiment with two Scots pine (*Pinus sylvestris*) placed in *PlaSTraGAS-gh7*. We focus on overall system setup, environmental controls, and $CH_4$ flux measurements. $CO_2$, $H_2O$, and VOC flux measurements follow routine dynamic chamber methods and are only discussed to the extent to which they are relevant for the overall system design.

## 2 Methods

### 2.1 System components

Both implementations of *PlaSTraGAS* consists of the following components (Fig. 1):

- Shoot and/or soil enclosure chambers (Section 2.1.1)

- A static () chamber module for trace gas flux measurements (2.1.2)

- Installations to ensure constant conditions during closed loop measurements (2.1.3)

- A dynamic (flow-through) chamber module for water, $CO_2$, and VOC flux measurements (2.1.4)

- Installations to flush inactive chambers with ambient air (2.1.5)

- A switching board that directs gas flows to and from the different chambers, measurement modes, and analysers (2.1.6)

- The control software and a central data recording system (2.1.7).

### 2.1.1 Plant and soil chambers

The *PlaSTraGAS-clim2* is equipped with one shoot chamber and one soil chamber, whereas *PlaSTraGAS-gh7* is equipped with with up to seven shoot chambers.


**Shoot chambers** (Fig. 2a) were custom built by Toivo Pohja Tmi (Juupajoki, Finland). The chambers' inner dimensions are 12 x 24 x 4 cm and each chamber encloses a volume of 1.15 L. The bottom and the rear plate of the chamber are constructed from aluminium, the other sides from UV-transparent acrylic glass covered with FEP tape on the inside of the chamber. UV transparency of the cover was confirmed by UV-VIS spectroscopy (Perkin-Elmer Lambda25; Fig. 2b). The connection between the removable cover and aluminium base of the chamber is sealed with a thin (1mm) foam gasket placed in a groove in the cover against the bottom of the base and in a groove in the aluminium rear plate against the rear end of the cover. The seal can be further improved by applying vacuum grease (Sigma Aldrich) to the gasket. The cover is attached to the base with eight screws; six against the bottom and two against the rear plate. To seal the opening for the shoot in the rear plate, the shoot is buffered with a pressure-sensitive adhesive (Blu-tack, Bostik S.A.) wrapped in PTFE tape at the chamber opening. The needles or leaves are held in place inside the chamber by means of a fishing line bed.

The bottom of each chamber is equipped with a Peltier cooling element. One fan is located inside each chamber, a second fan was placed outside below each Peltier element on a finned radiator. Each chamber is further equipped with a Pt 100 temperature probe (SKS Automaatio Oy) placed inside the chamber and a PAR sensor (Kipp & Zonen PQS1) placed on top of the chamber.

The **Soil chamber** consists of a custom built aluminium container of 40 x 40 x 30 cm (volume: 48L) with a cover made of the same acrylic glass as the shoot enclosure chamber (Toivo Pohja Tmi, Juupajoki, Finland). The container is flushed between the measurements by opening two circular vents on the sides of the container by means of flaps that are moved by pneumatically operated linear actuators. Fans are placed in front of the vents, as well as on the floor of the container. The stem of the plant being measured goes through an opening in the cover, which is then also sealed with the pressure-sensitive adhesive described above.

The total enclosed volumes in chamber, tubing, analyzer and pump were approximately 1.6 L (greenhouse system), 1.4 L (climate chamber system/shoot chamber), and 48.25 L (climate system/soil chamber).

### 2.1.2 Static chamber module for trace gas flux measurements

Trace gas fluxes are measured in a closed loop setup where air is recirculated between a shoot or soil chamber and one or more online gas analysers. In principle, any flow-through trace gas analyser or combination of analysers can be used with this setup given that it can (a) completely recirculate the analysed air into the enclosure chambers, and (b) the analyser does not emit the analysed trace gas or interfering volatile compounds (e.g. from pump membranes). At minimum one analyser capable of measuring $CO_2$ mixing ratios is required. Since our initial measurements were focused on $CH_4$ fluxes, we used a Picarro G2301 ($CH_4$ / $CO_2$ / $H_2O$) or a Picarro G2201i ($^{12/13}CH_4$ / $^{12/13}CO_2$ / $H_2O$) cavity ring-down spectroscopic analyser equipped with a KNF oil-free membrane vacuum pump. Analysers with low flow rates (e.g. the Picarro G2201i) require a bypass loop with a membrane pump (e.g. Nitto Kohki GMBH, model DP0140-A1111) to accelerate gas transport between a chamber and the analyser and thus reduce the lag between mixing ratio change occurring in the chamber and that being observed by the analyser. Analysers without an internal pump require an external pump to circulate the air between a chamber and the analyser.



### 2.1.3 Temperature, $CO_2$, and humidity control

**Temperature control** The enclosure temperature is controlled through the Peltier elements located beneath each shoot chamber. The Peltier element is activated when the temperature inside the shoot chamber exceeds ambient temperature (measured through an additional temperature sensor) by 2 °C and deactivated when the temperature inside the chamber drops 1 °C below the ambient temperature. Homogeneous temperature inside the chamber is ensured by the fan directing the air flow straight onto the area where the Peltier element is connected. This also minimises water condensation on the cooled area.

**Humidity control.** To avoid moisture build-up from transpired water during static chamber measurements, a membrane dryer (Nafion MD-050-12S-2) placed in the return line from the analyser to the soil and shoot chambers. The dryer is either flushed with dry air in counter-stream or evacuated with a vacuum pump (Gardner-Thomas, model 1410V).

**$CO_2$ control** To maintain $CO_2$ mixing ratios in the closed loop mode, $CO_2$ removed due to photosynthesis is replaced by $CO_2$ injections regulated by a mass flow control unit (MFC1; 0-50 mL/min, Bürkert GmbH, type 8715). We initially injected a 1% $CO_2$ in $N_2$ gas mixture utilising a PID algorithm to keep the $CO_2$ level stable. These injections, however, diluted the chamber air and decreased the trace gas mixing ratios. $CH_4$ mixing ratios in typical operation, for example, decreased by 100-300 ppb (5-15%) below ambient mixing ratios. Under these circumstances, small diffusion leaks can lead to an increase in the trace gas mixing ratios over time during chamber closures, which can be mistaken for shoot emissions.

After initial tests, we therefore changed the system to inject pure $CO_2$. In addition, we changed the injection algorithm to inject a fixed amount of $CO_2$ (0.14 mL, corresponding to approximately 400 ppm $CO_2$ in a shoot chamber) whenever the $CO_2$ mixing ratio falls below a configurable threshold value (set to 400 ppm). With this method, $CO_2$ injections have only a minimal effect on trace gas mixing ratios (e.g. <10 ppbv $CH_4$), and injections can be easily identified and corrected for. To facilitate rapid mixing of the injected $CO_2$ into the sample stream, we placed a hand-crafted flap in the fitting that connects that MFC to the main sample loop to force the sample air to flow through the throat of the MFC controlling the injections.

### 2.1.4 Dynamic chamber module for water, $CO_2$, and VOC flux measurements

Steady-state flow-through measurements are preferable over closed loop measurement when the gas fluxes can be quantified by measuring the difference in their mixing ratios in air entering and leaving the chamber. In our current setup, we use a dynamic chamber module to measure fluxes of $CO_2$ (photosynthesis rate), $H_2O$ (transpiration), and VOCs. To operate the chamber in dynamic chamber mode, pressured air (in-house) dried with a membrane drier (SMC, model IDG1-C06) is pushed into the enclosure chamber cell at a controlled flow rate. When VOC fluxes are measured, this air is further purified by a zero-air generator (HPZA 3500 220, Parker Balston) prior to use. The flow rate is controlled by a second mass flow control unit (MFC2, 0-1000 mL $min^{-1}$, Bürkert GmbH, type 8715) set to a constant flow rate (typically 850 ml $min^{-1}$). A bypass valve allows direct analysis of the air pushed into the enclosure chamber.

$CO_2$ and $H_2O$ mixing ratios during dynamic chamber closures are measured with a LiCor Li-850 gas analyser. A needle valve on the outlet of the Li-850 regulates the flow rate generated by its internal pump, such that the air flow pulled from the





enclosure chamber by the analyser(s) matches the air flow pushed into the chamber via MFC2. In addition, VOC mixing ratios
are measured by a proton transfer quadrupol mass spectrometer (PTR-QMS; Ionicon, Innsbruck, Austria).

### 2.1.5 Chamber flushing with ambient air

To keep the conditions in shoot chambers close to ambient between the flux measurements, shoot chambers are constantly
flushed with ambient air. Initially, this was achieved by placing an opening to ambient air into the gas lines just upstream the
shoot chambers. This inlet is protected by a check valve that only allows air inflow into the shoot chamber when the pressure
differential between the chamber and ambient air is more than -50 mbar. Shoot chambers were flushed by connecting separate
membrane pumps (Nitto Kohki GMBH, model DP0140-A1111) to each shoot chamber via the switching board. This way,
inactive shoot chambers are flushed with ambient air at a flow rate of 750-1000 $\mathrm{mL\,min^{-1}}$. Initial tests showed that the brushes
in these flush pumps burn out easily. We therefore changed the systems such that the chambers were flushed by pressing
pressurized air into the chambers. In this setup, the opening was moved downstream of the shoot chamber, and a check valve
was inverted, such that it allows air outflow but not inflow.

### 2.1.6 Switching board for connecting chambers to analysers

Each implementation contains a switching board that can connect each individual chamber to the static (trace gas analysers)
and dynamic ($CO_2$, $H_2O$, VOC analysers) chamber modules. In *PlaSTraGAS-clim2*, the switching board contains 6 electrically
operated 3-way solenoid valves (SMC VX3114K-01N-5G1-B) that direct the air flow from the desired outlets to the distinct
analysers. In the case of *PlaSTraGAS-gh7*, air streams are directed by a total of 36 solenoid shut-off valves (SMC VDW13-
5G-1-H-Q) located on 12 3-input-1-output manifolds (SMC VV2DW1-H03M5-F-Q). The inlet and outlet of each chamber can
be directed to three different lines (static chamber module, dynamic chamber module, chamber flushing). An additional 3-way
solenoid valves (SMC VX3114K-01N-5G1-B) is used to switch between chamber and bypass air in the dynamic chamber
module. Both setups allow the connection of one chamber to static chamber module while another chamber can be connected
to the dynamic chamber module; chambers not connected to either modules are operated in flush mode. In addition, each
system was equipped with a sampling inlet to analyze trace gas mixing ratios in ambient air.

### 2.1.7 Control software and data recording

Both measurement systems are operated by *Koppi*, a custom made software written in Python. The software allows for the
automatic switching between chambers and the measurement modes, regulates the $CO_2$ injections and Peltier coolers in re-
sponse to $CO_2$ and temperature data, and records the instrument configuration and all measurement data at 0.2 Hz frequency.
The volume of injected $CO_2$ is recorded at 0.1 Hz due to the slow response time of the MFC, and it interpolated to 0.2 Hz
frequency prior to data analysis. PTR-MS data is recorded separately and synchronized with the main measurement dataset
during data processing.



## 2.2 Data analysis and calculations

Data was processed in four steps from raw data to a time series of flux and auxiliary measurements that have been scaled to shoot measures where appropriate. All data processing was conducted in R version 3.6.3 (R Development Core Team, 2015).

In **step one**, raw data from the main operating software and auxiliary datasets (e.g., raw data recorded by internal dataloggers in the analysers) are imported, synchronized, and combined into a single data set. In addition, individual closures are identified with their start and end times, and the volume of injected $CO_2$ was interpolated to 0.2 Hz frequency.

**Step two** comprises corrections conducted at the raw data level. Most importantly, measurements in closed loop mode were corrected for the effects of $CO_2$ injections. For this, we modelled the mixing of $CO_2$ with chamber air after each injection. The model contained two elements, (a) mixing of injected $CO_2$ with air returning to the shoot chamber, and (b) mixing of air in the shoot chamber and air in the analyser loop.

For (a), mixing of air released by the MFC into the return air stream was described by an exponential decay function (eq. 1),

$$J_{effective}(t) = a \cdot f_{conv} \cdot \int_{t'=0}^{t} J_{MFC}(t') \cdot (1 - e^{\frac{(t-t')}{\tau}}) dt' \tag{1}$$

where $J_{effective}(t)$ stands for the effective flux at time point $t$ ($t = 0$ at the start of the modelled chamber closure), $a$ stands for an empirically fitted constant (2.5 for the test measurements presented in this study), $f_{conv}$ for the gas-specific conversion factor for thermal conductance based on mass flow measurements (0.7 for $CO_2$), $t'$ stands for a time point prior to $t$ during the same closure, $J_{MFC}(t')$ for the injection flux recorded by the mass flow control unit at that time, and $\tau$ for a fitted exponential
decay constant (90 sec) for the data analysed in this study) that is empirically fitted to describe the data in a given setup. After installing the metal flap at the tee-connector between MFC and return air flow, this component was not necessary anymore and instantaneous mixing (i.e., $Q_{effective}(t) = Q_{MFC}(t)$) could be assumed.

For (b), the system was conceptualized as the combination of the main chamber and a single tube with a volume equivalent to the total volume of all tubing and analyzers in the system. The tube was further modeled as consisting of n elements, each
holding a volume equivalent to the flow rate per time step (5 sec). At each time step, air was moved from tube element n to tube element n+1, the first tube element was filled with chamber air and the last tube element was emptied into the chamber. Measurements assumed to be conducted in tube element n/2-1, injections in tube element n/2. The flow rate was assumed based on the specifications of the analyser (400 mLmin$^{-1}$), while the number of tube elements (n=5) were fitted to the data.

We confirmed the validity of this model by applying it to $CO_2$ injections during nocturnal leakage tests (see below), when
$CO_2$ injections were conducted at set intervals rather than triggered by a mixing ratio threshold, and when the $CO_2$ emissions from foliar respiration were well characterized. We then calculated the corrected $CO_2$ and $CH_4$ mixing ratios according to equations 2 and 3,

$$[CO_2]_{corr}(t) = [CO_2]_{raw}(t) - [CO_2]_{inj}(t) \tag{2}$$



$$[CH_4]_{corr}(t) = \frac{[CH_4]_{raw}(t)}{1 - [CO_2]_{inj}(t)} \tag{3}$$

where $[CO_2]_{corr}(t)$ and $[CH_4]_{corr}(t)$ stand for the corrected dry $CO_2$ and $CH_4$ mixing ratios at time point $t$, $[CO_2]_{raw}(t)$ and $[CH_4]_{raw}(t)$ for the measured dry mixing ratios, and $[CO_2]_{inj}(t)$ for the mixing ratio of injected $CO_2$ at time point $t$.

This correction could be avoided when analysing data from *PlaSTraGAS-clim2* and when flux rates were sufficiently high to reduce the effective time of static chamber closures. Instead of correcting for the effect of $CO_2$injections, we identified such injections as local maximums of the $CO_2$mixingratio) were identified and time periods during which the $CO_2$ mixing
ratio change was affected by the injection were removed. We then treated the time periods between the injections as separate subclosures (see below). This approach was not possible in *PlaSTraGAS-gh7* due to the relatively long tube length between chambers and switching board (2x10m) which caused a relatively long delay until full mixing was reached after each injection.

Other corrections during this processing step included converting $CO_2$ mixing ratios conducted by the Li-850 to mixing ratios in dry air. In the test experiments presented herein, we also had to apply a a 6-minute running average filter on $CO_2$
mixing ratios to remove an oscillation of measured values due to an instrument malfunction. Raw data from PAR measurements (in mV) was converted to PAR (in $\mu mol\,m^{-2}\,sec^{-1}$) using the calibration equations provided by the manufacturer.

**Step three** consisted of data reduction by calculations of derived values for each closure. For static chamber closures, this was conducted differently depending on whether gas mixing ratios had to be corrected for $CO_2$ injections or not. For data measured by *PlaSTraGAS-gh7* with $CO_2$ injection corrected gas mixing ratios, we calculated the slope of each measured
gas's mixing ratio over time ($dC/dt$) as the simple linear regression between mixing ratio and time. For data measured by *PlaSTraGAS-clim2* with identified sub-closures between injections, a a function with a single slope ($dC/dt$) for all sub-closures of but distinct intercepts for each sub-closure was fitted onto each main closure.

For dynamic chamber closures, the mixing ratio in air leaving the chamber ($C_{out}$) was calculated as the average mixing ratio measured from 180 sec after closure start to 60 sec before closure end. Similarly, the mixing ratio in air entering the chamber
($C_{in}$) was calculated as the average mixing ratio from 180 sec after closure start to 60 sec before closure during bypass periods. Auxiliary measurements (PAR, temperature) were averaged over the entire closure time.

**Step four** consisted of calculating gas fluxes and normalizing them to sample size (e.g., foliage dry weight or leaf area). For static chamber closures, $dC/dt$ was then used to calculate the flux rate per leaf area ($Q_A$) or leaf dry weight ($Q_m$) according to eqs. 4 and 5,

$$Q_A = \frac{dC}{dt} \cdot \frac{1}{A} \cdot \frac{V}{V_{mol}} = \frac{dC}{dt} \cdot \frac{V}{A} \cdot \frac{p}{R \cdot T} \tag{4}$$

$$Q_m = \frac{dC}{dt} \cdot \frac{1}{m} \cdot \frac{V}{V_{mol}} = \frac{dC}{dt} \cdot \frac{V}{m} \cdot \frac{p}{R \cdot T} \tag{5}$$

where $A$ and $m$ stands the leaf area and leaf dry weight of the enclosed branch, $V$ for the chamber volume including analyser loop, and $V_{mol}$ molar volume, which is calculated from pressure $p$, temperature $T$, and ideal gas constant $R$.





For dynamic chamber closures, $CO_2$, $H_2O$, and VOC fluxes were calculated as described in eqs. 6 and 7,

$$Q_A = \frac{flowrate}{A} \cdot (C_{out} - C_{in}) \tag{6}$$

$$Q_m = \frac{flowrate}{m} \cdot (C_{out} - C_{in}) \tag{7}$$

where *flow rate* stands for the air flow rate rate ($850\,\mathrm{ml\,min^{-1}}$) and $C_{out}$ and $C_{in}$ stand for the measured gas's mixing ratio in air leaving the chamber and air entering the chamber, respectively.

## 2.3 System validation tests

### 2.3.1 Leakage

The leakage rate ($L$) of each chamber was quantified by injecting $CO_2$ to until its mixing ratio reached approximately 3000 ppmv, and monitoring the decline in the $CO_2$ mixing ratio due to gas exchange between the chamber and ambient air. These measurements were conducted automatically once per night for each chamber, and corrected for nighttime respiration rates ($Resp$) measured prior to each leak test.

$$\frac{dC_{chamber}(t)}{dt} = L \cdot (C_{ambient} - C_{chamber(t)}) + Resp \tag{8}$$

During initial tests, we also quantified $L$ taking advantage of the initial measurements where a 1% $CO_2$ in $N_2$ mixture was used to replace the photosynthesized $CO_2$. These injections decreased the mixing ratio inside the shoot chamber ($C_{chamber}$) by 5–10 % (to 1.8–1.9 ppmv), while the $C_{ambient}$ remained constant ($\sim 2.0$ ppm). We used these variations in chamber $CH_4$ mixing ratios to calculate $L$ as the regression between the change in the mixing ratio of $CH_4$ over time ($dC/dt$) and $C_{ambient} - C_{chamber}$, assuming that any $CH_4$ exchange between shoot and chamber air is not affected by $C_{chamber}$.

### 2.3.2 Blank tests

Our initial tests were focused on the ability of the chamber systems to accurately measure shoot $CH_4$ emissions. We therefore evaluated the system blank for $CH_4$ exchange from shoot chamber, but not for other greenhouse gases or the soil chamber. To quantify the system blank, all openings of the shoot chamber were closed and the systems were operated in the same way as for plant shoot measurements. These measurements were either conducted before and after each experiment (*PlaSTraGAS-clim2*) or during the experiment with chambers left empty for blank control (*PlaSTraGAS-gh7*). We furthermore calculated the system detection limits for individual chamber closures as equal to three times the standard deviation of the blank measurements.





### 2.3.3 Test measurements with Scot's pine shoots

Test measurements were conducted with the *PlaSTraGAS-gh7* system and a two year old Scots pine (*Pinus sylvestris L.*)
sapling. The sapling was obtained from a commercial grower in Fall 2019, potted in a 20 L pot, and stored outdoors in
the University of Helsinki's Viikki greenhouse facility over the winter. In late January 2020, the tree was transferred into a
greenhouse compartment and allowed to acclimatize for three weeks prior to the measurement campaign (Feb 22-25). The
ambient temperature in the compartment was between 15 and 18 °C during nighttime and warmed to 22 to 32 °C during
daytime, depending on weather conditions. The trees were watered weekly, and received additional light from a high pressure
sodium lamp resulting in 250-400 $\mu mol\,m^{-2}\,sec^{-1}$ photosynthesis active radiation (PAR). In addition, we placed 6 UV-A
lamps (Q-lab UVA-340) approximately 20cm above the measured shoot to stimulate aerobic $CH_4$ production. Both PAR and
UV lighting followed 12h day/night cycles (7am to 7pm).

We installed a total of four automated shoot chambers into the system. Chambers 1 and 4 were kept empty as blank controls,
while chambers 2 and 3 were placed on separate branches of the sapling. As exposure to sunlight was low, we decided not to
cool the chambers with the Peltier cooling system to keep the experiment more simple. The system was programmed to place
connect each shoot chamber to a Picarro G2301 analysis via static chamber module for 24 minutes followed by measuring
ambient air for 3 minutes. To explore the effects of $CO_2$ injections on $CH_4$ flux measurements, $CO_2$ injections were de-
activated during every second closure cycle (Fig. 5). This cycle was restarted every two hours. Only 'daytime' measurements
(i.e., artificial lighting on; 7am-7pm) were included in the presented data, while the results of the temporal trends (e.g., diurnal
cycles) will be published separately. We obtained a total of 25–26 measurements per chamber.

Concurrent with each static chamber closure, a different chamber was connected for 12 minutes to the Li-850 and PTR-QMS
analysers, followed by analyzing the in-going pressurized air for 15 minutes (Fig. 6). For simplicity, only three molecular mass-
to-charge ratios were monitored: 33 (methanol), 59 (acetone), and 137 (monoterpenes). The PTR-QMS was calibrated with a
gas standard containing methanol, acetone, $\alpha$-pinene, as well as other VOCs not measured in this study. Data processing was
conducted as described previously (**?**).

After the experiment, the enclosed shoots were cut from the tree and the (projected) needle leaf area was quantified by
scanning an subset of the needles and scaling to the whole branch by weight. The needle dry weight was quantified after drying
for 48h at 80 C.

We state our main measurement result - $CH_4$ fluxes - as mean and 95% confidence interval because our focus here the overall
uncertainty associated with the average flux found in these measurement. Results from auxiliary measurements — temperature,
PAR, $CO_2$ and water fluxes — are presented as means and standard deviation, because we primarily present these results to
document the conditions under which the trace gas measurements were conducted.

The measurements $CH_4$ fluxes were close to the detection limit and measurements of both empty and pine shoot chambers
had long-tailed distributions (i.e., contained likely outliers). To test for differences in apparent $CH_4$ fluxes between the shoot
chambers, we therefore applied the non-parametric Kruskal-Wallis test and Nunn post-hoc tests as normal distribution could
not be assumed.





## 2.4 Assessment of measurement uncertainties

We identified three potential sources of inaccuracy in the measurements; chamber leakage, $CO_2$ injection modelling, and spectral interference by volatile organic compounds. We assessed the impact of these potential errors by propagating the uncertainty caused by these processes onto measured $CH_4$ fluxes. All estimates were scaled based on chamber closure times (24 min), leaf areas (0.02 $m^2$), and foliar dry weights (3g) in this study. To evaluate the impact of gas exchange with ambient air due to chamber leakage, we assumed a mixing ratio difference between chamber and ambient air of 10 ppbv and a chamber leakage rate $L=1.5\%$. For the effect of inaccuracy of the $CO_2$ injection model, we assumed a 250 ppmv inaccuracy in the mixing ratio of $CO_2$ in the injection model and a $CH_4$ mixing ratio of 2 ppmv. Finally, to evaluate the potential effect for spectral interferences by co-emitted VOCs, we assumed methanol, acetone, and monoterpene emission rates based on the average emission rates found in this study (1.54, 2.55, and 2.33 $\mathrm{nmol\,g^{-1}\,d.w.\,h^{-1}}$. respectively). Based on these emission rates, we estimated the mixing ratio of plant-emitted methanol, acetone, and monoterpenes reached at the end of static chamber closures as 28.5, 47.4, and 43.3 ppbv, respectively. We note that this approach likely overestimates the final VOC mixing ratios as increasing headspace VOC mixing ratios often lead to a decrease in emission rates and even net-uptake of VOCs by foliage (Cojocariu et al., 2004; Cappellin et al., 2017). Nevertheless, we consider them a good conservative estimate for assessing the potential impact of VOC emissions on $CH_4$ flux measurements. We converted these VOC mixing ratios to apparent $CH_4$ mixing ratios based on our recent quantification of upper limits to the spectral interference of various VOC in methane mixing ratio measurements with the Picarro G2301 and other methane analysers (Kohl et al., 2019a), using conservative uncertainty limits ($\pm0.4$ ppbv apparent $CH_4$ $\mathrm{ppmv^{-1}}$methanol and $\pm0.2$ ppbv apparent $CH_4$ $\mathrm{ppmv^{-1}}$monoterpenes). Since the spectral interference of acetone was not quantified by Kohl et al. (2019a), we applied the higher values value derived from methanol.

## 3 Results and Discussion

### 3.1 Temperature control

Initial tests of *PlaSTraGAS-clim2* showed that cooling was not necessary as the enclosure chambers do not warm significantly compared to the ambient (cabinet) temperature due to the low thermal energy emitted by the LED based lighting system. In a test consisting of 1311 closures with pine seedlings in the chamber, the mean difference in temperatures between lights on and lights off was found to be $1.06 \pm 0.03$ °C, and the median change in chamber temperature during measurement was $6\mathrm{x}10^{-6}$ $\mathrm{C\,s^{-1}}$.

Temperature measurements with *PlaSTraGAS-gh7* conducted in August 2019 showed that uncooled shoot chambers can heat to 10 °C and more above ambient temperature during summer conditions in northern Europe. Cooling allowed us to keep the difference between ambient and chamber temperature below 2 °C (Fig 4). In the test measurements with pine shoots conducted in the greenhouse in February 2020, uncooled chambers warmed to 3–4 °C above the ambient temperature when the room lighting was on (Tab. 5), indicating that moderate cooling is required for experiments under greenhouse conditions even during winter months.





### 3.2 H$_2$O control

The membrane dryer was capable of reducing the moisture in an empty shoot chamber connected to the static chamber module to <10% relative humidity within <5 minutes (Fig 5a). During the measurements with pine shoots in the chamber, the membrane drier removed sufficient water from the chamber to prevent condensation of transpired water in the system and hold the relative humidity in the shoot chamber between 40 and 50%.

### 3.3 CO$_2$ control

Photosynthesis by the enclosed pine shoots depleted CO$_2$ in the enclosed volume to <100 ppm within 2-3 minutes. In the test experiments with pine shoots, an injection corresponding to approx. 400 ppm CO$_2$ was triggered once every 10 minutes. These injections allowed to sustain the CO$_2$ between 400 and 700 ppm (Fig. 7) for extended periods of time (tested for up to 2 hours). While maintaining more constant CO$_2$ mixing ratios is possible with this system, pulsed injections make it easier to correct trace gas mixing ratios for dilution by the injected CO$_2$.

To evaluate the performance of the CO$_2$ injection model, we evaluated 20 leak test measurements. In these nighttime measurement, shoot CO$_2$ emissions and leakage were well characterized, such that the effect of CO$_2$ injections on measured CO$_2$ mixing ratios could be studied in isolation of other processes (Fig. 3). The model generally predicted CO$_2$ mixing ratios within 250 ppmv. Assuming a CH$_4$ mixing ratio of 2 ppmv, the propagated error of CH$_4$ mixing ratios due to this uncertainty is <0.5 ppbv.

### 3.4 Chamber leakage


Initial tests showed relatively high leakage rates of up to 1–2% per minute (2). Over time, we made improvements to the chamber seal (e.g., application of vacuum grease to contact surfaces, testing seal with a hand held pressure meter while closing the shoot chamber.) This resulted in lower leakage rates, <0.15% min$^{-1}$ in the climate chamber system and <0.5 % min$^{-1}$ in the greenhouse system. This leakage rate has negligible effects on flux measurements when the analyte gas's initial initial

mixing ratio in the shoot chamber is close to its mixing ratio in the ambient air surrounding the shoot chamber (cabinet air in the case of the climate chamber). It is currently not common to report leakage rates in static chamber studies, and we are therefore unable to compare these rate literature values. However, we hope that this reporting becomes more common to allow for such a comparison in the future.

Chamber leakage becomes a more serious issue when the analyte gas's mixing ratio inside the chamber ($C_c$) differs signifi-

cantly from its mixing ratio in ambient air ($C_a$). This is relevant in two cases: (a) when the CO$_2$ injections strongly dilute the analyte gas inside the shoot chamber, or (b) when the analyte gas's mixing ratio inside the climate chamber cabinet increase due to strong emissions from the plant or soil. We observed, for example, elevated CH$_4$ mixing ratios in the cabinet air when a *Betula nana* plant growing in water saturated peat was placed in the cabinet. In these cases, an apparent flux of $L \cdot (C_a - C_i)$ occurs, and needs to be corrected for during data analysis.



### 3.5 System blank and method detection limit

Average system blanks, that is, the apparent $CH_4$ flux in an empty control chamber, were <0.3 nmol h$^{-1}$ in both systems, corresponding to a mixing ratio change of <1.8 ppbv $CH_4$ during a 24 minute chamber closure. Method detection limits (MDL) for $CH_4$ emissions from plant shoots were <0.15 nmol g$^{-1}$ d.w. h$^{-1}$ in the climate chamber system and <1.5 nmol g$^{-1}$ d.w. h$^{-1}$ in the greenhouse system (assuming 3 g d.w. foliar biomass per chamber; Table 2). This method detection limit is defined for a single closure measurement and further decreases with $\sqrt{n}$ in the case of repeated measurements. It is thus easy to reach a MDL well below reported plant methane emissions rates (e.g., 0.75 - 55 nmol g$^{-1}$ d.w. h$^{-1}$; (Keppler et al., 2006)).

### 3.6 Test measurements with Scot's pine shoots

#### 3.6.1 Auxiliary measurements

The two enclosed shoots contained needles with a total dry weight of 2.61 and 3.92 g dry weight and leaf areas of 0.019 and 0.027 m$^2$, respectively. During the included test (i.e., daytime) measurements, average temperature and PAR were 24.1C (SD 3.4; range 16.5 to 31.8) and 328 $\mu$mol m$^{-2}$sec$^{-1}$ (SD 104; range 62 to 620). As mentioned above, the measured temperatures inside the shoot chambers were higher than ambient temperature, on average by 3.3 °C (SD 1.8; range -2.6 to 10.3). Temperature and PAR values of individual chambers are summarized in Table 5.

The average measured $CO_2$ mixing ratio (1 SD) of air entering the shoot enclosure in dynamic chamber mode was 384.8±5.5 ppmv (Fig. 8). After passing through empty chambers, $CO_2$ mixing ratios were on average slightly elevated (390.6±5.8 and 391.1±5.7 ppmv, respectively), whereas $CO_2$ was significantly depleted after air passed through shoot chambers (295.9±18.3 and 308.6±17.3 ppmv). The average carbon uptake by pine shoots, calculated as the difference between shoot and empty chamber, were 7.63±1.39 and 9.16±1.93 mmol $CO_2$ m$^{-2}$ leaf area h$^{-1}$ (Table 3).

The average measured absolute humidity air entering the chamber was -0.048±0.005 %; the slightly negative values likely resulted from a miss-calibration of the instrument (Fig. 8c). The humidity after passing through empty chambers was slightly elevated (measured values -0.030±0.007 and -0.031 %±0.005, respectively), and significantly elevated after air passed through shoot chambers (1.105±0.232, 1.1064±0.230 %, respectively). The average transpiration by pine shoots, calculated as the difference between shoot and empty chamber, were 1.24±0.26 and 0.90±0.18 mol m$^{-2}$ leaf area h$^{-1}$ (Table 3).

The mixing ratios of three volatile compounds (classes) monitored in this study — methanol, acetone, and monoterpenes — in the air entering the chambers were 1.82±0.01, 0.10±0.01, and 0.20±0.04 ppbv, respectively (Fig. 8e,g,i). The mixing ratio of these compounds in air leaving empty chambers were 4.12 and 4.54; 1.43 and 1.64; and 0.47 and 0.48 ppbv; their mixing ratios in air leaving chambers with pine shoots were 6.38 and 6.95; 4.13 and 7.05; and 2.56 and 5.96 ppbv. The emission rates of methanol, acetone, and monoterpenes, calculated as the difference between shoot and empty chamber, were therefore 0.11±0.05 and 0.21±0.11; 0.29 ± 0.20 and 0.44±0.25; and 0.24±0.10 and 0.44±0.23 nmol m$^{-2}$ leaf area h$^{-1}$ (Table 3). These emission rates are comparable to field measurements (e.g. Tarvainen et al., 2005).



### 3.6.2 Methane flux measurements

The apparent $CH_4$ emission rates and their 95% confidence intervals were $0.700\pm0.137$ and $1.106\pm0.170$ nmol h$^{-1}$ in chambers with pine shoots, and $0.279 \pm0.134$ and $0.445\pm0.111$ nmol h$^{-1}$ in empty chambers (Fig 7a). Apparent emission rates in chambers with pine shoots were significantly different from the empty chambers and from each other, whereas fluxes from the

two empty chambers were not significantly different from each other (Kruskal-Wallis $\chi^2$ = 52.8, p<0.001). Apparent $CH_4$ production rates of pine shoots were significantly lower for closures with $CO_2$ injections compare to closures without injections (Fig 7b), representing the dilution of $CH_4$ by the injected $CO_2$. However, apparent $CH_4$ production rates were near identical to those measured from the same shoot without $CO_2$ injections when $CH_4$ mixing ratios were corrected for this dilution. This demonstrates the correction of CH4 mixing ratios successfully compensated for effects of $CO_2$ injections. It also indicates that

there was not short-term response of $CH_4$ emissions rates to the inhibition of $CO_2$ fixation rates due to low $CO_2$ mixing ratios.

Scaled and blank-corrected $CH_4$ fluxes were $0.130\pm0.062$ and $0.190\pm0.047$ nmol g$^{-1}$ foliar d.w. h$^{-1}$ or $18.1\pm8.7$ and $28.0\pm7.2$ nmol g$^{-1}$ m$^{-1}$ leaf area h$^{-1}$ (Table 3). These values are approximately five-fold below the lowest values reported by Keppler et al. (2006) for living plant tissues, but 5-10 times higher than fluxes measured from shoots of mature Scots pine trees (Machacova et al., 2016) (median 3.13 nmol m$^{-2}$ leaf area h$^{-1}$). A number of reasons may have led to these relatively

low emissions rates compared to experiments by Keppler et al. (2006), including the timing of our measurements during the early growing season and the relatively low PAR irradiation provided in our experiments. Conversely, the higher emissions in our experiment compared to field measurements of the same species might have resulted form the augmented UVA irradiation or the fact that Machacova et al. (2016) conducted measurements during cloudy days only to avoid the overheating of their manual shoot enclosure. Regardless, these measurements demonstrate that our system is capable of detecting and quantifying

$CH_4$ emissions at or below the levels observed in many laboratory and field conditions.

Our evaluation of potential sources of measurement uncertainty (Table 4) indicated that chamber leakage was the main source of error in $CH_4$ flux measurements. Measurement errors due to leakage were of a similar size as the observed fluxes, which explains the relatively large variability of empty chamber $CH_4$ fluxes. Chamber leakage, however, equally affected chambers with pine shoots and empty chambers (Fig. 7a) and should therefore not lead to biased results if measurements from a sufficient

number of chamber closures are averaged and corrected for apparent fluxes observed in empty chambers. Furthermore, these results also indicate that better chamber tightness will lead to an improvement in detection limit of the method. In contrast, the effects of inaccuracies in the $CO_2$ injection model and spectral interferences by VOC were five and ten times smaller than the observed fluxes, respectively, indicating that there mechanisms had only minor impacts upon measurement accuracy.

## 4   Conclusions

We developed an automated system to measure trace gas fluxes from plant shoots and other plant compartments while controlling the temperature, $CO_2$ mixing ratio, and humidity in the plant chamber. Initial tests demonstrated that the system can detect $CH_4$ fluxes at the scale reported for plant shoots. The system also allows the monitoring water, $CO_2$, and VOC fluxes. It is built in a modular way that is easy to customize and/or expand to different chamber types. We have constructed two implementations



of this setup that are designed to measure trace gas fluxes from a single plant under controlled environmental conditions in a
growth chamber, and from multiple plants in a greenhouse compartment. Future development will aim to adapt the system to
allow its deployment under field conditions.

*Code and data availability.* Raw measurement data and the analysis script are available at Zenodo (10.5281/zenodo.4609836). The software
used to operate both systems is available online at https://bitbucket.org/makoskinen/koppismear.

*Author contributions.* MPi developed the initial concept. MK, LK, and MPi set out the design goals. MK, TP, LK, and KR came up with
engineering solutions. TP was the main engineer responsible for building the systems with help from MK, LK, and KR. MK wrote the control
software for the systems, LK the data analysis pipeline. LK, MK, MPa, and AZ analysed data from validation experiments. ST conducted the
validation experiments with help from MK, LK, TP, and MPa. LK wrote the first draft of the manuscript with inputs from MK. All co-authors
contributed to the final manuscript.

*Competing interests.* The authors declare no competing interests.

*Acknowledgements.* We would like to thanks Juho Aalto and Heikki Laakso for advice while designing the system, and Olli-Pekka Tikkasalo
for help with parameterizing the $CO_2$ injection model.

This project was funded by the European Research Council (ERC) under the European Union's Horizon 2020 research and innova-
tion program (grant agreement number 757695) and the Academy of Finland (grant numbers 319329 and 2884941). LK received a Marie
Skłodowska-Curie Actions Fellowship from the European Union's Horizon 2020 research and innovation program (grant agreement number
843511); MK a postdoctoral fellowship from the Maj and Tor Nessling foundation.



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

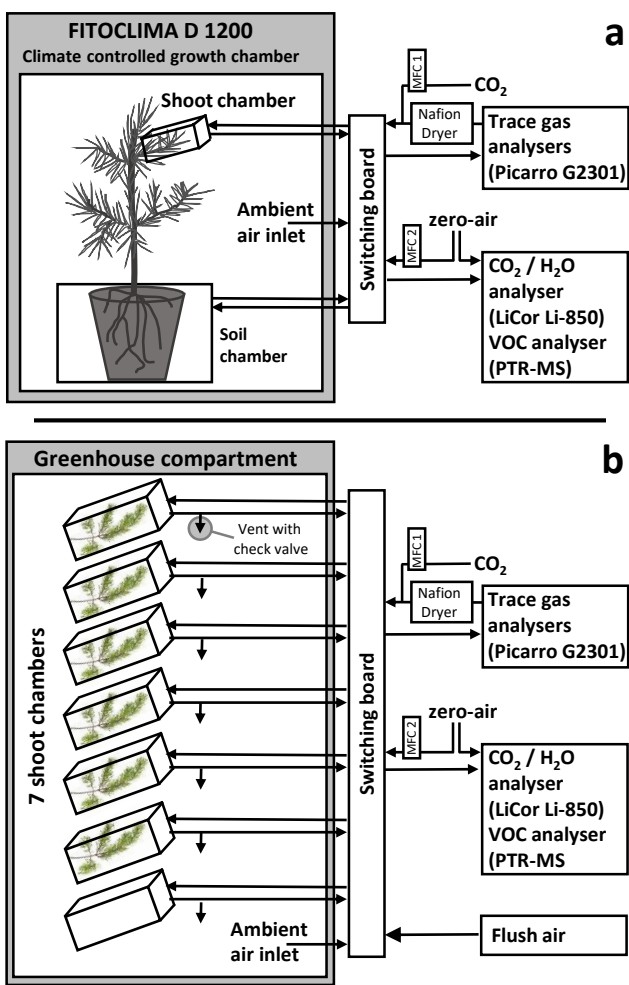

**Figure 1.** Schematic of *PlaSTraGAS*, a measurement system to quantify trace gas exchange at plant shoots. We constructed two implementations of this system: *PlaSTraGAS-clim2* consists of one shoot and one soil chamber places inside a climate-controlled cabinet (**a**), whereas *PlaSTraGAS-gh7* consists of seven shoot chambers and is located in a greenhouse compartment (**b**). Both systems allow measuring trace gas fluxes in a static chamber module and major gases and volatiles in a dynamic chamber module.





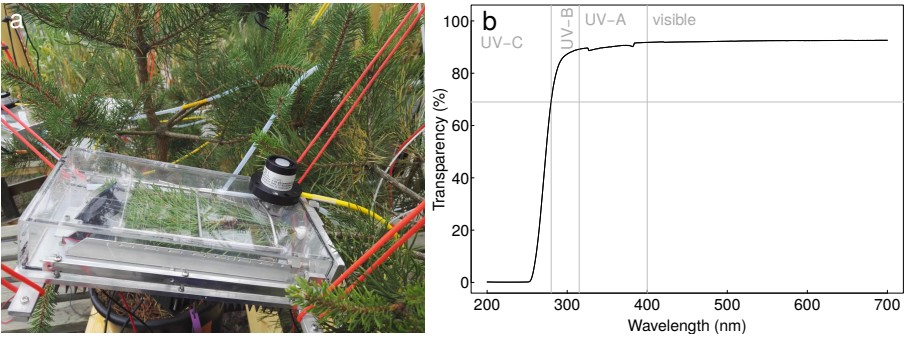

**Figure 2.** Picture of the shoot enclosures used for *PlaSTraGAS* (**a**) and the UV-VIS transmission spectrum of the transparent chamber cover (**b**).

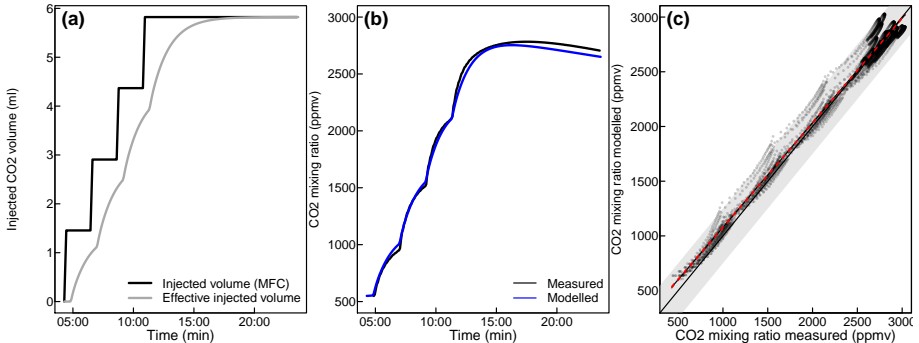

**Figure 3.** Injected $CO_2$ volume (quantified by mass flow controller) and effective injected $CO_2$ volume during a leak check test of an empty chamber (**a**). The effective volume takes into account delays due to mixing in different parts of the system. Further, measured and modelled $CO_2$ mixing ratios during the same closure (**b**). Moreover, comparison of measured against modelled $CO_2$ mixing ratios during all 20 leak checks performed during the test measurements (**c**). The solid black line indicates the equal measured and modelled mixing ratios and the dashed red line a linear regression between measured and modelled data. The shaded grey area indicates that the difference between measured and modelled values was less than 250 ppmv.



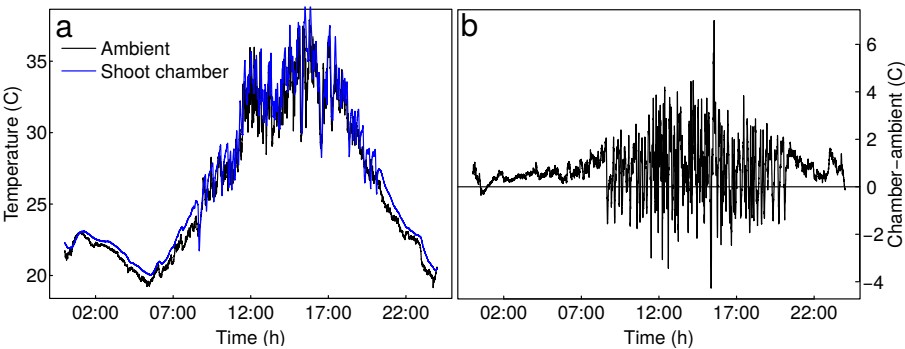

**Figure 4.** Example of the temperature control used with *PlaSTraGAS-gh7*: ambient temperature and temperature in a shoot chamber a the greenhouse compartment (**a**) and the temperature difference between shoot chamber and ambient air (**b**). Data was measured on Aug 1 2019.



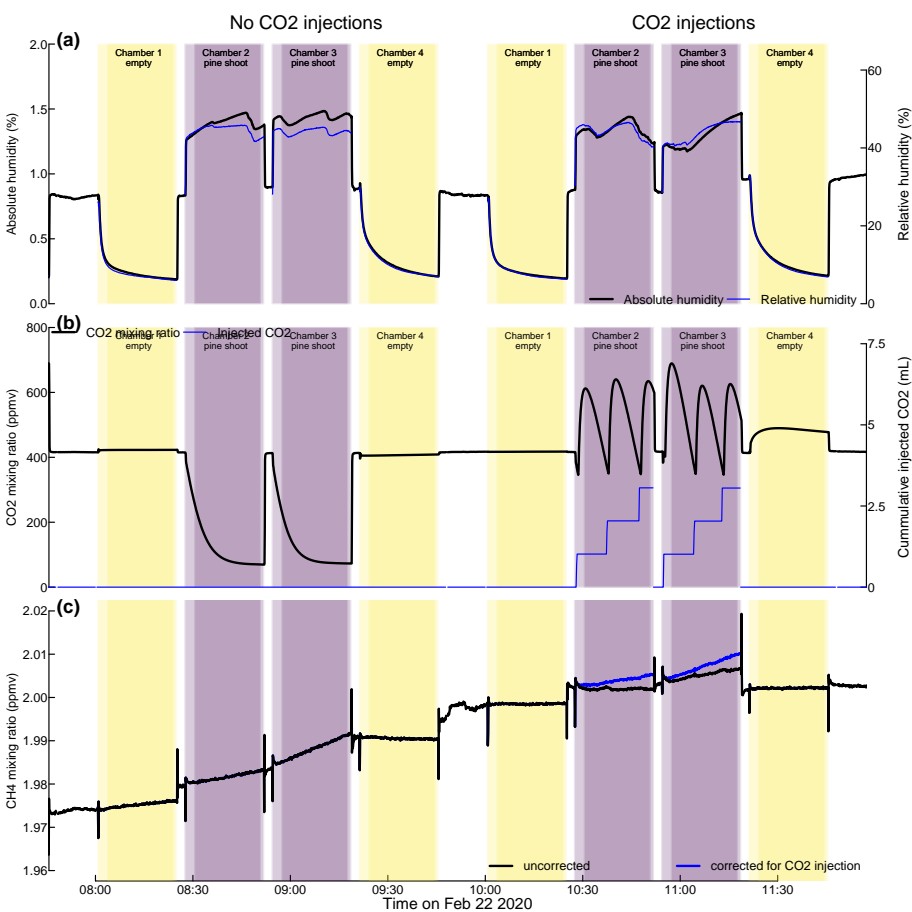

**Figure 5.** Mixing ratios of water (**a**), $CO_2$ (**b**), and methane (**c**) during static chamber closures of four shoot chambers in the greenhouse system. Chambers 2 and 3 each contained a shoot of a two year old pine sapling, chambers 1 and 4 were kept empty as blank controls. The figure depicts two sets of chamber closures that were conducted without and with $CO_2$ injections to compensate for plant $CO_2$ uptake, respectively. Black lines in panels (**a**) to (**c**) represent the measured mixing ratios of water (**a**), $CO_2$ (**b**), and methane (**c**), respectively. The blue line in panel **b** indicates the cumulative amount of $CO_2$ injected since the beginning of the chamber closure, expressed as the equivalent mixing ratio in the chamber (right hand axis). The blue line in **c** indicates the methane mixing ratio after correcting for dilution by the injected $CO_2$ (see text). Shaded areas indicate times when chambers with or without shoots were connected to the static chamber module with darker colours indicating times used to calculate flux rates. The analyzer was connected to an ambient air inlet between these closure times. The depicted data was measured on Feb 22 2020.

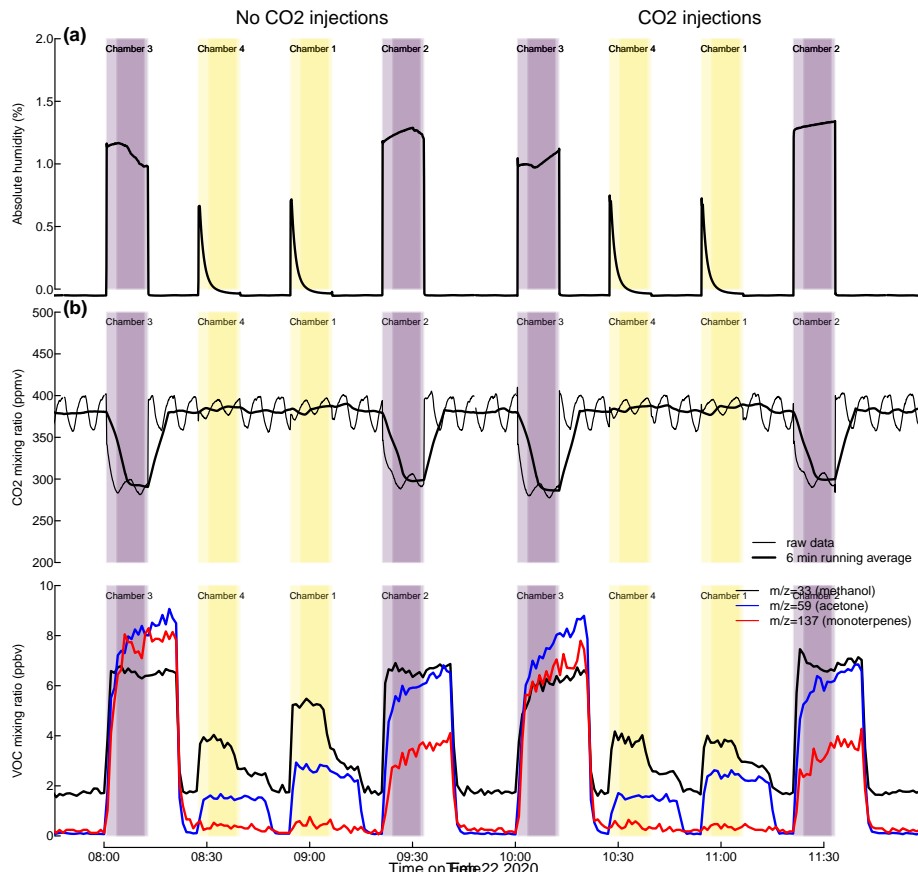

**Figure 6.** Mixing ratios of water (**a**), $CO_2$ (**b**), and volatile organic compounds (**c**) during dynamic chamber closures of four shoot chambers in the greenhouse system. Chambers 2 and 3 each contained a shoot of a two year old pine sapling, chambers 1 and 4 were kept empty as blank controls. Black lines represent the measured mixing ratios of water (**a**) and $CO_2$ (**b**). In panel (**b**), the thin black line represents the raw measured $CO_2$ mixing ratio, while the thick black line represents its six minute running average, calculated to compensate for an oscillation in the analyser signal. Shaded areas indicate periods where chamber air was analyzed, with darker colours indicating time periods used to calculate $C_{out}$, non-shaded areas periods when the ingoing air was measured bypassing the chamber. The depicted data was measured on Feb 22 2020.

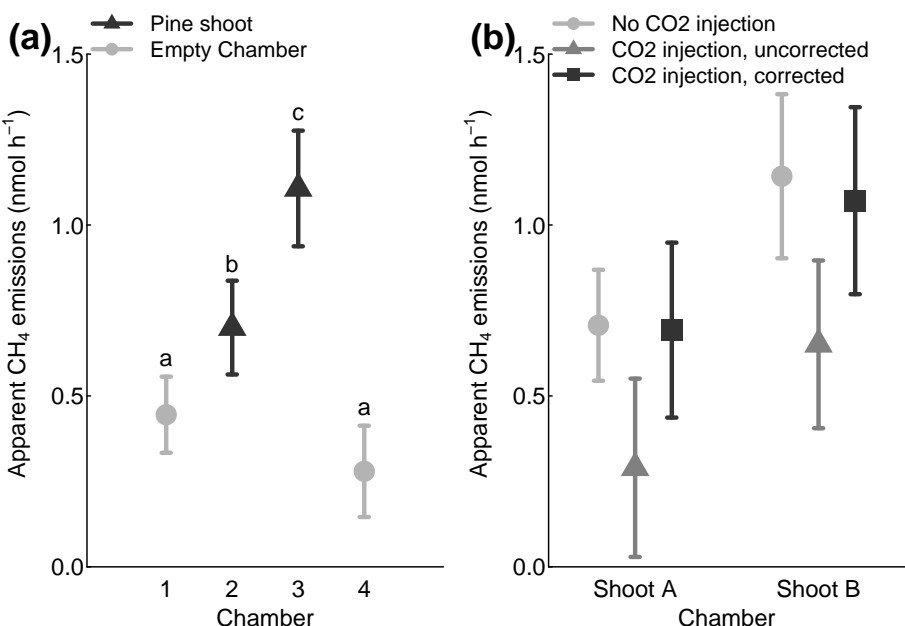

**Figure 7.** Observed apparent $CH_4$ fluxes in two empty shoot chambers and two shoot chambers with pine shoots (**a**). Furthermore, comparison of $CH_4$ fluxes during chamber closures without and with $CO_2$ injections (**b**). Only daytime (illuminated) measurements are included in the figure. Error bars indicate 95% confidence intervals.

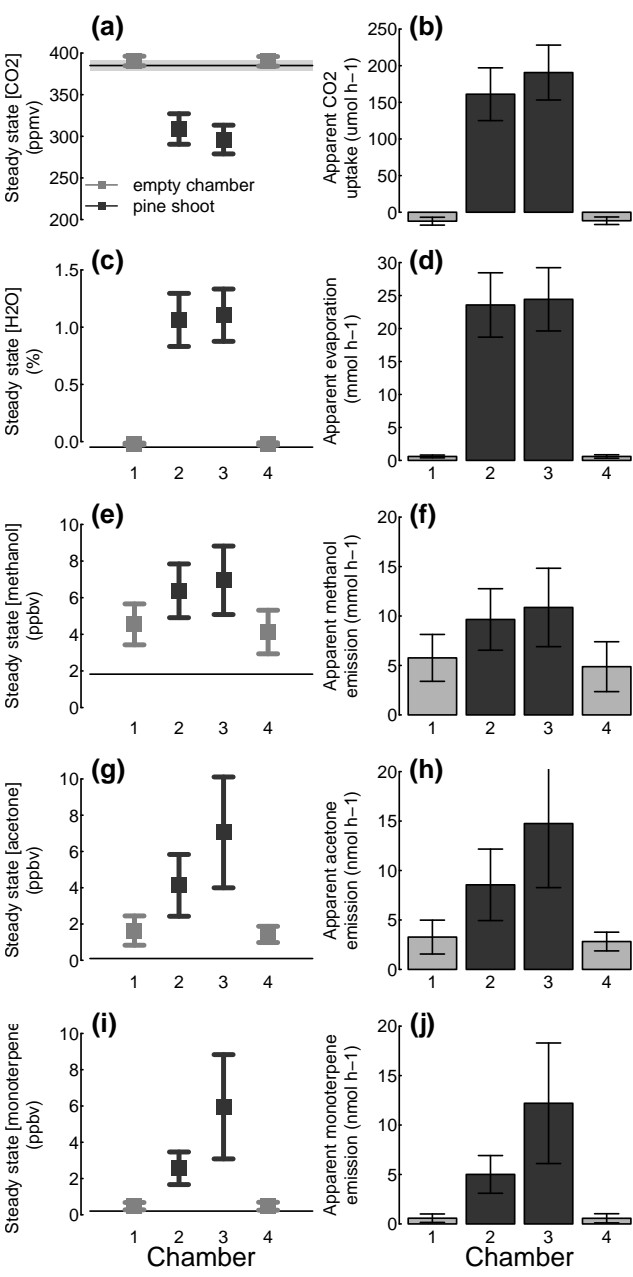

**Figure 8.** Observed steady-state mixing ratio of $CO_2$ (**a**) water (**c**), and VOCs (**e,g,i**) in outgoing air ($C_{out}$) during dynamic chamber measurements of two empty chambers (grey) and two chambers with pine shoots (black). The mixing ratio of $CO_2$ water in ingoing air ($C_{in}$) are indicated by the horizontal lines in each plot. Further, apparent $CO_2$ uptake (**b**), transpiration (**d**), and VOC emission (**f,h,j**) rates calculated from these mixing ratios. Error bars and the shaded area around the horizontal lines indicate one standard deviation.



**Table 1.** Shoot characteristics.

| Measure | Unit | Chamber 1 (empty) | Chamber 2 (shoot A) | Chamber 3 (shoot B) | Chamber 4b (empty) |
|---|---|---|---|---|---|
| Foliage dry weight | g | | 2.61 | 3.92 | |
| Projected leaf area | $m^2$ | | 0.0188 | 0.0266 | |
| Temperature (daytime, SD) | C | 24.1±3.1 | 23.7±2.9 | 24.1±4.3 | 23.0±3.1 |
| Photosynthetic active radiation (daytime, SD) | $\mu mol\,m^{-2}\,sec^{-1}$ | 341±67 | 246±73 | 360±141 | 322±86 |

**Table 2.** Observed chamber leakage rates, system blanks, and method detection limits

| Date | System | Application | Leakage rate $L^a$ (% $min^{-1}$) | System blank[b] (nmol $h^{-1}$; SD) | Method detection limit[c] (nmol $g^{-1}$ d.w. $h^{-1}$) |
|---|---|---|---|---|---|
| May 2019 | Climate chamber | pine shoot | 0.338 ± 0.005 | | |
| Mar 2020 | Climate chamber | 2 empty chambers | 0.138± 0.006 | | |
| Oct 2020 | Climate chamber | 1 empty chamber | 0.102± 0.013 | 0.257± 0.137 | 0.137 |
| Feb 2020 | Greenhouse | 2 empty, 2 pine shoots | 1.276±0.296 | 0.172± 1.196 | 1.196 |
| Feb 2020 | Greenhouse | 1 empty, 3 pine shoots | 1.724±1.048 | | |
| Mar 2020 | Greenhouse | 1 empty, 6 pine shoots | 0.314±0.171 | | |
| Mar 2020 | Greenhouse | 1 empty, 6 pine shoots | 0.400±0.206 | 0.290±1.362 | 1.362 |

[a]Diffusive air exchange between chamber and ambient air. Measured by comparing the nighttime CO2 trend at ambient mixing ratios and after injecting $CO_2$ to a mixing ratio of 2000-3000 ppmv.

[b]Flux observed in empty control chambers

[c]Method detection limit for a single measurement, defined as three times the standard deviation of the system blank, and normlized to the foliage dry weight of a typical shoot (3 g). The detection limit for repeated measurements decreases with $\sqrt{n}$.



**Table 3.** Shoot fluxes measured in this study scaled to foliar dry weight and leaf area after after subtracting empty chamber fluxes. All uncertainties include the uncertainties in shoot and blank (empty chamber) measurements.

| Flux | Unit | Shoot A | Shoot B |
|---|---|---|---|
| $CH_4$ emissions | (nmol $g^{-1}$ d.w. $h^{-1}$; CI) | 0.130±0.062 | 0.190±0.047 |
| | (nmol $m^{-2}$ $h^{-1}$; CI) | 18.1±8.7 | 28.0±7.2 |
| $CO_2$ uptake | (mol $g^{-2}$ d.w. $h^{-1}$; SD) | 0.066±0.014 | 0.052±0.010 |
| | (mol $m^{-2}$ $h^{-1}$; SD) | 9.20±1.94 | 7.62±1.42 |
| Transpiration | (mol $g^{-2}$ d.w. $h^{-1}$; SD) | 0.0088±0.0019 | 0.0061±0.0012 |
| | (mol $m^{-2}$ $h^{-1}$; SD) | 1.24±0.26 | 0.90±0.18 |
| Methanol emission | (nmol $g^{-2}$ d.w. $h^{-1}$; SD) | 1.66±1.52 | 1.41±1.19 |
| | (nmol $m^{-2}$ $h^{-1}$; SD) | 230±210 | 208±175 |
| Acetone emission | (nmol $g^{-2}$ d.w. $h^{-1}$; SD) | 2.11±1.48 | 2.99±1.69 |
| | (nmol $g^{-2}$ $h^{-1}$; SD) | 293±204 | 440±249 |
| Monoterpene emission | (nmol $g^{-2}$ d.w. $h^{-1}$; SD) | 1.70±0.75 | 2.96±1.56 |
| | (nmol $m^{-2}$ $h^{-1}$; SD) | 236±104 | 437±230 |

**Table 4.** Additional sources of uncertainty in $CH_4$ fluxes

| Source | $CH_4$ mixing ratio uncertainty (ppbv) | $CH_4$ flux uncertainty (mol $h^{-1}$) | (mol $g^{-1}$ d.w. $h^{-1}$) | (mol $m^{-2}$ $h^{-1}$) |
|---|---|---|---|---|
| Chamber leakage | ±<3.6 | ±<0.58 | ±<0.19 | ±<29 |
| CO2 injection model | ±<0.50 | ±<0.081 | ±<0.027 | ±<4.0 |
| Methanol spectral interference | ±<0.11 | ±<0.018 | ±<0.0061 | ±<0.92 |
| Acetone spectral interference | ±<0.19 | ±<0.031 | ±<0.0102 | ±<1.53 |
| Monoterpene spectral interference | ±<0.09 | ±<0.014 | ±<0.0047 | ±<0.67 |