# Peer review of "An automated system for trace gas flux measurements from plant foliage and other plant compartments"

_Atmospheric Measurement Techniques, 2021_

## Author Comment (AC1)

*Reviewer #1*

*Traditionally static chamber might largely bias the flux measurements of trace gases on plant shoots due to plant physiological activity. This study developed a novel system, PlasTraGAS, for continuous and automated measurements of trace gas exchange at plant shoots by regulating temperature, humidity, and CO2 concentrations in the shoot enclosure. This system holds the potential for providing insights into the role of plant foliage in the global budgets of trace gases.*

*This is a good work.*

**We thank Reviewer #1 for their positive feedback and we have further improved our manuscript following the their suggestions.**

*However, I have the following concerns.*

*As we know, leaf chamber in LiCor series instruments is used for measuring photosynthesis. Please provide a discussion on difference between your new system and LiCor series instruments. What is advantage of your new system?*

Licor manufactures a series of instruments (LI-6800 and its predecessor LI-6400XT) optimized for measuring leaf-level $CO_2$ and water fluxes in a dynamic chamber setup. When combined with external analysers, these systems can also be used for dynamic-chamber measurements of other species. They are optimized for a quick installation on individual leaves and cannot provide sufficiently leak-tight closures for static chamber measurements. These systems can therefore not be used to measure $CH_4$ and $N_2O$ fluxes at the rates at which they occur at typical plant shoots.

**We added the following wording to the Introduction**: "[…] as currently commercially available leaf-level trace gas exchange measurement systems (e.g. Licor Li-6800) are limited to dynamic chamber measurements and provide insufficient leak tightness for static chamber measurements" (L48-49).

*When an instrument is expensive and complicated, it is hard to be widely applied in the field. Can your system be widely used in forests in nature?*

Briefly, we are currently working on making *PlaSTraGAS* field portable. At this point, we are aiming at developing a version that can be installed for whole-growth-season deployment at long-term research sites. The system is indeed burdensome and requires sufficient infrastructure (power, pressurized air supply, $CO_2$ in gas cylinders) and not likely to result in a system that can be carried to remote field sites by the user. However, we think that this is acceptable given that this is the first prototype of a system capable of measuring shoot $CH_4/N_2O$ fluxes.

**We added the following point to the conclusions section:** "Future development will aim to adapt the system to allow its deployment under field conditions, e.g., at long term monitoring sites" (L439-440).

*I find your system is tested in pine saplings. Obviously, in nature most of tree stems are much larger than your shoots. Can your system be extended to large stems of trees in forests?*

Yes, the system can be combined with any type of measurement chamber. However, we deliberately de-emphasize this point as such systems already exist (Barba et al. 2019b) and do not require the degree of temperature, moisture, and $CO_2$ regulation we implemented for shoot measurements.

**We added the following sentences to the Methods section:** "*PlaSTraGAS* follows a modular design, such that different types of static chambers can be connected to the measurement system. This allows the system to be adopted to plants with distinct shoot geometries (e.g., coniferous versus deciduous trees), and to include other surfaces (e.g. tree stems)" (L79-81).

> *Thus, I recommend a revision with additional discussion.*

We hope that we were able to address the reviewers concerns in the revised manuscript.

---

## Author Comment (AC2)

*Reviewer #2*

*General comments*

*Kohl and colleagues developed an automated plant chamber to measure trace gas and VOC fluxes from plant shoots. The system includes cooling elements, removal of transpiration water and an automated system to replace fixed CO2. With this system it should be possible to relate trace gas exchange of plant shoots –related to leaf area- to environmental conditions and plant physiological patterns. In their manuscript they introduce the chamber technique itself and provide substantial results from initial tests a 'Transpiration rate (dynamic chamber mode) per dry weight')nd validation experiments.*

*In general, the manuscript is very well written and easy to follow. The design of the chamber is well thought out and will certainly improve the current technique to measure trace gas emissions from plants in the field. Also the test measurements appear to have been well carried out and the results are convincing. The section about measurement uncertainties including interferences with VOC is adequate. Although often discussed, only a couple of experiments take interferences with VOCs into account.*

**We thank Reviewer #2 for their positive feedback and their suggestions how to improve the manuscript.**

*What about the leakage associated with the shoot entrance? Could this be a problem by causing different leakages when changing branches between measurements, thereby leading to different leakages?*

We agree that the shoot entrance is definitely a weakest point for the tightness of the shoot enclosure. We address this by conducting nightly measurements to quantify the leak rate in each individual chamber, and to measure ambient concentrations, so at the very least the leakage rate can be taken into account during flux calculations. This will become particularly important during the future development of the system for field measurements (where e.g. wind forces can weaken the sealing over time).

**We added the following sentences to the Results and Discussion section:** "It is, however, possible that during longer experiments the sealing around the shoot inlet deteriorates due to physical stress, leading to larger leakage in shoot with tree branches compared to empty controls. It is therefore important to continuously monitor the tightness of each chamber throughout such experiments, as is currently done with automatic nightly measurements" (L424-429).

*Did you observe any artefacts due to pressure effects in the system?*

Pressure artifacts associated with the beginning and end of chamber closures can be seen e.g. as 'spikes' in Fig 5c. However, we excluded the time periods immediately after closing the chambers during which these artifacts occur are excluded from data analysis. The chambers themselves are vented to the atmosphere and should not undergo pressure changes >50 mbar. This is also important as we learned that any significant pressure difference to ambient air leads to the development of leaks in the chamber sealing.

**We added the following sentence to methods section** "In both cases, data measured during the the first 180 sec after the closure start and the last 60 sec before the end of the closure were removed to exclude minor artifacts resulting from pressure effects (visible e.g. in Fig. 5c) and the mixing of distinct air volumes" (L222-224).

*Moreover, I've got a remark regarding plant physiology. Gas exchange depends on stomatal conductance. Would it be possible to calculate stomatal conductance of leafs with the parameters given by your chamber system? If so, it might be possible to relate stomatal conductance to trace gas fluxes. It could be interesting to see how fluxes change depending on stomatal conductance/humidity/light etc.*

Yes, that is the intention behind measuring $CO_2$ and $H_2O$ fluxes concurrently with the trace gas fluxes. **We added stomatal conductance values to Table 3 and the Results and Discussion section** (L393-394)**.** We also added the formula used to calculate stomatal conductance to methods section (L245-253).

*The manuscript is of high quality and deserves publication in Atmospheric Measurement Techniques. Therefore, I recommend publication of this manuscript with minor revisions.*

**Thanks again for your positive response to our work!**

*Minor comments*

*Fig 5+6 Please revise figure label (x and y scale + legend), the letters are too small or -in case of the legend- overlap.*

**Have changed the figures accordingly.**